# Ursolic acid regulates gut microbiota and corrects the imbalance of Th17/Treg cells in T1DM rats

**Weiwei Chen[1], Yingying Yu[2,3], Yang Liu[1], ChaoJie Song[3], HuanHuan Chen[3], Cong Tang[3], Yu Song[3], Xiaoli Zhang[3]***

1 Traditional Chinese Medicine (Zhong Jing) School, Henan University of Chinese Medicine, Zhengzhou, Henan Province, China, 2 The Second Affiliated Hospital of Luohe Medical College, Luohe, Henan Province, China, 3 Henan University of Chinese Medicine School of Medicine, Zhengzhou, Henan Province, China

* zxl7666@163.com

**Data Availability Statement:** All relevant data are within the paper and its Supporting Information files.

## Abstract

Ursolic acid (UA), a natural pentacyclic triterpenoid obtained from fruit and several traditional Chinese medicinal plants, exhibits anti-inflammatory and hypoglycemic properties. However, its protective effects against type 1 diabetes mellitus (T1DM) have not been explored. In this study, streptozotocin-induced T1DM rat models were established and treated with UA for six weeks. T1DM rats treated with UA were used to observe the effects of UA on body weight and fasting blood glucose (FBG) levels. Pathological changes in the pancreas were observed using immunohistochemical staining. The gut microbiota distribution was measured using 16S rDNA high-throughput sequencing. The proportions of Th17 and Treg cells were examined using flow cytometry. Protein and mRNA expression of molecules involved in Th17/Treg cell differentiation were assessed by quantitative real-time PCR and western blotting. The correlation between gut microbiota and Th17/Treg cell differentiation in T1DM was analyzed using redundancy analysis (RDA) analysis. Compared with the model group, FBG levels declined, and the progressive destruction of pancreatic β cells was alleviated. The diversity and uniformity of gut microbiota in T1DM rats treated with UA increased significantly. Interestingly, the Th17/Treg cell differentiation imbalance was corrected and positively correlated with the expression of Foxp3 and IL-10, and negatively correlated with the expression of RORγt, IL-17A, and TNF-α. These findings suggest that UA can lower FBG levels in T1DM rats, delay the progressive destruction of pancreatic β-cells, and modulate gut microbiota homeostasis and immune function in streptozotocin-induced T1DM rats.

## 1. Introduction

Type 1 diabetes mellitus (T1DM) is a chronic metabolic disease in which the autoimmune system progressively damages pancreatic β-cells, leading to insulin deficiency. Hypoglycemia, ketoacidosis, and microvascular diseases are potentially life-threatening complications. T1DM

**Funding:** This work was supported by the Henan Province Key University Project: Study on the hypoglycemic mechanism of ursolic acid based on the regulation of Th17/Treg cell balance by gut microbiota(20B310005) ,the Henan Province Science and Technology Research Project: The mechanism of ursolic acid in alleviating type 1 diabetes based on the regulation of RORγt/Foxp3 signaling pathway by gut microbiota (212102311081) and the Henan Province Key University Project: Study on the mechanism of ursolic acid on IFN-γ and IL-4 cytokine signal transduction in diabetes(172102310091).

**Competing interests:** The authors have declared that no competing interests exist.

is one of the most frequent metabolic diseases in adolescents and children, with increasing incidence and prevalence each year [1, 2]. The number of adolescents (<20years old) living with T1DM worldwide is close to 1.1 million [3]. The number of patients with T1DM is alarmingly high and is now recognized as resulting from the interaction of environmental factors with genetic inheritance, microorganisms, and the immune system [4]. Current drugs for T1DM treatment are far from optimal; therefore, alternative effective and safe drugs need to be developed.

Although genetic inheritance is an important factor affecting susceptibility to T1DM, the gut microbiota and balance between Th17 and regulatory T(Treg) cells play an important role in the pathogenesis of T1DM [5–8]. Specifically, the gut microbiome has been proven to be a critical factor in T1DM based on its ability to change intestinal permeability and regulate mucosal immunity [9]. Gut microbiota dysbiosis in patients with T1DM is usually characterized by low microbial diversity [10], loss of beneficial microflora, and an increased proportion of pathogenic bacteria [11, 12]. Liu *et al.* reported that fecal microbial dysbiosis occurs in T1DM [13]. A case-control study suggested that children with T1DM had a significant increase in the abundance of *Bacteroidetes*, a significant decrease in the number of *Firmicutes* and *Actinobacteria*, and a lower *Firmicutes* to *Bacteroidetes* ratio [14]. The significant changes in the number of bacterial communities and their ratios could be responsible for glycemic levels and altered gut permeability. Other evidence from animal experiments implies that propolis treatment can enhance insulin sensitivity and repair intestinal mucosal damage in diabetic rats by increasing short-chain fatty acids [15]. In mice and humans, gut microbiota plays an important role in inducing tolerance to autoantigens in the gut mucosa, which has been suggested to modulate the balance between pro-inflammatory and regulatory immune responses. In addition, Kim *et al.* reported that the microbial combination could be effective in autoimmune diabetes in non-obese diabetic (NOD) mice by altering gut permeability, increasing the generation of gut-homing Treg cells, and reducing Th1 polarization [16].

The balance between Th17 and Treg cells is crucial for autoimmune responses and the pathogenesis of metabolic syndrome. A recent study reported an imbalance in Th17/Treg cell differentiation in patients with T1DM, which manifested as an increase in Th17 cells and a concomitant decrease in Treg cells [17]. Inhibiting the development of Th17 cells and promoting Treg cell production could mitigate the severity of insulitis in female NOD mice [18]. As a CD4$^+$ T cell subset, Th17 cells are proinflammatory and can secrete IL-17 specifically under the regulation of nuclear factor ROR-γt. IL-17 can promote the secretion of proinflammatory cytokines, such as IL-6 and TNF-α, by antigen-presenting cells and participate in and expand the host inflammatory response. Treg cells are anti-inflammatory and are essential for the maintenance of immunological tolerance. Treg cells primarily suppress immune cell activity by secreting anti-inflammatory cytokines, such as TGF-β and IL-10. As the master transcription factor of Treg cells, Foxp3 is indispensable for Treg cells to develop and maintain their specific functions. Furthermore, Foxp3 inhibits Th17 cell differentiation but promotes Treg cell differentiation by establishing interactions with RORγt [19]. The above evidence suggests that modulation of the gut microbiota and restoration of the balance between Th17 cells and Treg may be a promising T1DM treatment strategy.

Ursolic acid (UA, 3β-hydroxyurs-12-en-28-OIC acid) is a pentacyclic triterpenoid derived from medicinal plants such as *Salvia officinalis*, *Lavandula angustifolia*, *Melissa officinalis*, *Ocimum basilicum*, *Origanum majorana*, *and Satureja montana* [20]. It has various pharmacological activities, such as hypoglycemic, antioxidant, anti-inflammatory, antitumor, and neuroprotective functions [21–23]. UA exhibits anti-cancer effects by inducing tumor cell cycle arrest and apoptosis. It also suppresses tumor transformation and inhibits tumor angiogenesis and tumorsphere formation. Moreover, the neuroprotective effects of UA could be

associated with its ability to stimulate cellular antioxidant defense systems and downregulate pro-inflammatory pathways in neurodegeneration models *in vitro* and *in vivo* [20, 24, 25]. The hypoglycemic effect of ursolic acid is generally recognized. Tang *et al.* [26] found that UA could decrease oxidative stress in pancreatic tissue to ensure the regeneration of pancreatic β-cells, and therefore, pancreatic insulin. It has also been suggested that UA exhibits potential anti-diabetic effects, as it protects islet cells from streptozotocin (STZ)-induced damage and modulates blood glucose levels [27]. Despite the well-documented excellent hypoglycemic activity of UA, little is known about its potential pharmacological effects on T1DM, Th17/Treg cell balance and gut microbiota.

Therefore, in this study, we examined whether UA exerted protective effects in rats with STZ-induced T1DM. We also investigated the effect of UA on gut microbial composition and Th17/Treg balance.

## 2. Materials and methods

### 2.1 Ethics statement

All experimental procedures were performed at the Experimental Animal Center under protocol number DWLL201903001. The experiments were conducted under the supervision of the Experimental Animal Welfare and Ethics Committee of Henan University of Chinese Medicine. All surgeries were performed under sodium pentobarbital anesthesia, and efforts were made to minimize suffering.

### 2.2 Animal studies

Eighty male Wistar rats (6 weeks, 170–180 g) were purchased from Jinan Pengyue Company. All experimental procedures were approved by the Ethics Review Committee of Henan University of Chinese Medicine. The animals were kept in a specific pathogen-free (SPF) environment with free access to food and water (20–26°C, 50–55% relative humidity, and a 12 h light/dark cycle). After one week of acclimatization, the T1DM model was established by a single intraperitoneal injection of 55 mg/kg (bw) streptozotocin (STZ; Solarbio Technology Co. Ltd., Beijing, China). The rats in the control group were intraperitoneally injected with the same dose of citric acid buffer. Rats were identified as having T1DM based on their fasting blood glucose (FBG) levels (three average value higher than 16.7 mmol/L) in blood drawn from the tail one week after STZ treatment. One week after STZ injection, 62 rats developed hyperglycemia, with blood glucose levels > 16.7 mmol/L.

The rats were randomly divided into six groups (n = 10-12/group): blank control (untreated) (Control), model (Model), metformin (Xinyi Pharmaceutical Factory Co. LTD., Shanghai, China) (MET), low-dose [25 mg/kg (bw)/day] UA (Bide Medical Technology Co. LTD., Shanghai, China) (UA-L), middle-dose [50 mg/kg (bw)/day] UA (UA-M), and high-dose [100 mg/kg (bw)/day] UA (UA-H). Rats in the control and model groups were treated with an equal volume of sodium carboxymethyl cellulose buffer; rats in the MET group were treated with 100 mg/kg (bw)/day MET by gavage; rats in the UA-L, UA-M, and UA-H groups were treated with corresponding doses of UA by gavage. All the rats received food and drinking water throughout the experimental period. The body weight of each rat was recorded twice weekly, and FBG was measured weekly after 12 h of fasting, using blood collected from the caudal vein.

After six weeks of treatment, all rats were anesthetized with sodium pentobarbital and blood samples were collected by abdominal aorta puncture. Fecal samples from the colon were collected and stored at -80°C for 16S ribosomal DNA sequencing. Pancreatic tissues were immersed in 4% paraformaldehyde and embedded in paraffin for immunohistochemical

staining. Spleen tissues were divided into two parts: one part was placed in PBS solution for timely flow cytometry analysis, and the other part was stored at -80˚C.

## 2.3 Immunohistochemical staining

Pancreatic tissues were fixed in 4% paraformaldehyde, embedded in paraffin, and sectioned into 3-μm-thick slices on slides. After deparaffinization and rehydration, paraffin sections were heated in a microwave oven for antigen unmasking. Then, the sections were stained with insulin antibodies (Servicebio, Wuhan, China) at 4˚C overnight and subsequently incubated with horseradish peroxidase-conjugated secondary antibodies (Servicebio, Wuhan, China) at room temperature for 50 min. Finally, the sections were stained with hematoxylin and imaged using a light microscope (Olympus, Tokyo, Japan).

## 2.4 Fecal DNA extraction and 16S rDNA gene sequencing

Fecal samples were collected from the colon and stored at -80˚C until DNA extraction. DNA was isolated from the samples using an E.Z.N.A.®soil DNA kit (Omega Bio-tek, USA). PCR amplification of the V3-V4 region of the bacterial 16S rRNA gene was performed using the 338 F/806R primer set (338 F: 5′–ACTCCTACGGGAGGCAGCAG–3′; 806R: 5′–GGAC–TACHVGGGTWTCTAAT–3′) incorporating sample barcode sequences. After purification and quantification of the PCR products, sequence libraries were generated and index codes were added following the manufacturer's recommendations. The libraries were then sequenced on a MiSeq platform using a paired-end (2× 300) sequencing strategy. Sequences were quality-filtered using QIIME and merged using FLASH. The optimized sequences were clustered into operational taxonomic units (OTUs) using the UPARSE pipeline with 97% sequence identity.

## 2.5 Flow cytometry

An appropriate amount of spleen tissue was used to obtain single-cell suspensions. The cells were harvested and stained with fluorescein isothiocyanate (FITC)-labeled anti-human CD4 and phycoerythrin (PE)-labeled anti-human CD25 for surface staining. After fixation and permeabilization, cells were stained with the intracellular cytokine antibodies allophycocyanin (APC)-labeled anti-Foxp3 and (PE)-labeled anti-IL-17A. The proportions of Treg and Th17 cells were analyzed using a flow cytometer (BD Biosciences), and the data were analyzed using Flow Jo 10.0 (Tree Star, Ashland, OR, USA).

## 2.6 Enzyme-linked immunosorbent assay (ELISA)

IL-17A,IL-10, and TNF-α levels were determined using commercial rat ELISA kits (Boster Biological Technology Co. Ltd., Wuhan, China), according to the manufacturer's instructions.

## 2.7 Total RNA extraction and quantitative real-time PCR (qPCR)

Total RNA was extracted from splenic tissues using an animal RNA isolation kit with a spin column (Beyotime Biotechnology Co. Ltd., Shanghai, China). RNA was reverse-transcribed into cDNA using a First Strand cDNA Synthesis Kit (Beyotime Biotechnology Co. Ltd., Shanghai, China). SYBR™ Green Master Mix was used to perform qPCR on the 7500 Real-Time PCR System (Applied Biosystems, Waltham, MA, USA). The primers used to amplify RORγt,Foxp3 and GAPDH are listed in Table 1. Data were analyzed using the $2^{-\Delta\Delta Ct}$ method, with GAPDH as the reference marker.

**Table 1. List of primers used for quantitative real-time PCR.**

| Genes | Primers |
|---|---|
| GAPDH-F | 5′-ACAGCAACAGGGTGGTGGAC-3′ |
| GAPDH-R | 5′-TTTGAGGGTGCAGCGAACTT-3′ |
| RORγt-F | 5′-CCTCTACACGGCCTTGGTTC-3′ |
| RORγt-R | 5′-ATGATGAAAGGCCAGCTCCAA-3′ |
| Foxp3-F | 5′-CCATAATATGCGGCCCCCTT-3′ |
| Foxp3-R | 5′-GCGGGGTGGTTTCTGAAGTA-3′ |

## 2.8 Western blotting

Total protein extracted from splenic tissues was denatured for western blot analysis. After determining the protein concentration, the lysates were loaded, separated on sodium dodecyl sulfate-polyacrylamide gel electrophoresis gels, and transferred to PVDF membranes. The membranes were blocked in PBS with Tween-20 detergent (TBST) and 5% skimmed milk for 1 h at 37°C and incubated with specific primary antibodies against RORγt and Foxp3 over-night at 4°C. After incubation for 1 h at 37°C with secondary antibody, the results were detected using Alpha Innotech alphaEaseFC, with β-actin used as the internal reference.

## 2.9 Redundancy analysis (RDA) analysis

The subset of environmental factors was screened by the bioenv function, and then RDA was performed on the sample flora and a subset of environmental factors, including Th17 cells, Treg cells, Th17/Treg cell ratio, blood glucose and body weight. After obtaining the most significant environmental factors affecting the distribution of sample flora, a correlation heatmap chart of the R value, *P* value, sample bacteria, and environmental factors was drawn to visually display the correlation between bacteria and environmental factors using the Pheatmap software.

2.10 Statistical analysis

Statistical analyses were performed using SPSS version 22.0 (IBM Corp., Armonk, NY, USA), and graphs were plotted using the GraphPad Prism 7.0 software (GraphPad Software, San Diego, CA, USA). R results are expressed as mean ± standard deviation (SD). Statistical comparisons among multiple groups were analyzed for significance using one-way ANOVA followed by the least significant difference test (LSD) or Dunnett's T3 post hoc analysis. Spearman correlation analysis of the gut microbiota and environmental factors was performed using R and P values. $P < 0.05$ was considered statistically significant at $P < 0.05$.

# 3. Results

## 3.1 UA treatment alleviated pancreatic β cell injury and reduced FBG level in T1DM rats

To assess the effects of UA on T1DM rats, we monitored the body weight and FBG levels of rats in all groups. As expected, after the intraperitoneal injection of STZ, the body weight of T1DM rats began to decrease significantly, whereas it continued to increase in the control group. Compared with the model group, the body weights of rats in the UA-M, UA-H, and MET groups were notably increased after 6 weeks of treatment ($P < 0.01$) (Fig 1A). One week after STZ injection, all rats in the model and treatment groups developed higher FBG levels than those in the control group ($P < 0.01$). Treatment with both UA and MET led to a lesser increase in FBG levels compared to T1DM rats ($P < 0.01$) (Fig 1B). Notably, histological

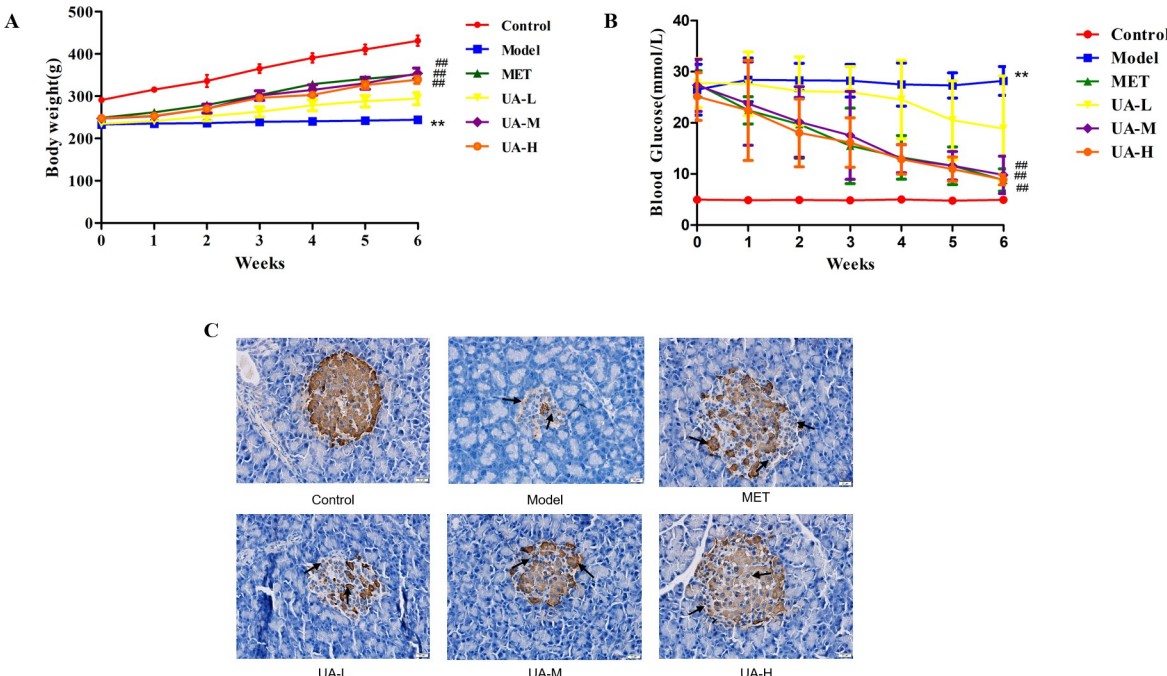

**Fig 1. UA treatment alleviated pancreatic β cells injury and reduced FBG level in T1DM rats.** Rats were divided into six groups: blank controls (Control), T1DM rats (Model), T1DM rats treated with metformin (MET), T1DM rats treated with low-dose [25 mg/kg (bw)/day] UA(UA-L), middle-dose [50 mg/kg (bw)/day] UA(UA-M) and high-dose [100 mg/kg (bw)/day] UA(UA-H). (A) Body weight change: Each group of rats was weighed weekly during the observation period (n = 10–12). (B) Fasting blood glucose levels (n = 10–12). (C) Histological changes in pancreatic islets and insulin secretion (shown by black arrows) were detected by IHC staining(original magnification ×400; scale bar, 200 μm). Data are expressed as mean ± SD. $^{**}P<0.01$ versus the control, $^{*}P<0.05$ versus the control; $^{##}P<0.01$ versus Model,$^{#}P<0.05$ versus Model.

analysis showed that the model group displayed distinctly abnormal islet structures compared with those of the control group; they appeared as small islets with irregular morphology and indistinct boundaries with the surrounding tissue, inhomogeneous islet cells, cytoplasmic vacuolation, fat vacuoles, and infiltration of inflammatory cells into islets. However, UA-treated rats had a significantly larger islet surface area and a lower number of insulin-positive cells in the islets than the rats in the model group (Fig 1C). Together, these results indicated that UA treatment significantly ameliorated STZ-induced T1DM.

## 3.2 UA treatment altered the specific microbiota population in T1DM rats

To investigate the role of the gut microbiota in T1DM and the effect of UA, we analyzed the relative abundance, diversity, and structure of the gut microbiota in stool samples using 16S rDNA sequencing. A total of 1231141 high-quality reads from 24 samples were generated. Based on a 97% similarity, 817 OTUs were obtained, which were divided into 12 phyla, 19 classes, 32 orders, 62 families, and 166 genera. As can be seen from the Pan/Core curve (Fig 2A), as the sample size of each group increased to four, the increase and decrease in the total OTU number of each group tended to be gentle, indicating that the amount of detection for each group was sufficient.

The results of alpha diversity analysis, reflecting the diversity and abundance of colony species, are shown in Fig 2B. Our results showed that Shannon and Simpson values were significantly different among the control, model, MET, and UA-M groups, suggesting that MET and UA can increase the diversity of the gut microbiota in T1DM rats. However, there were no

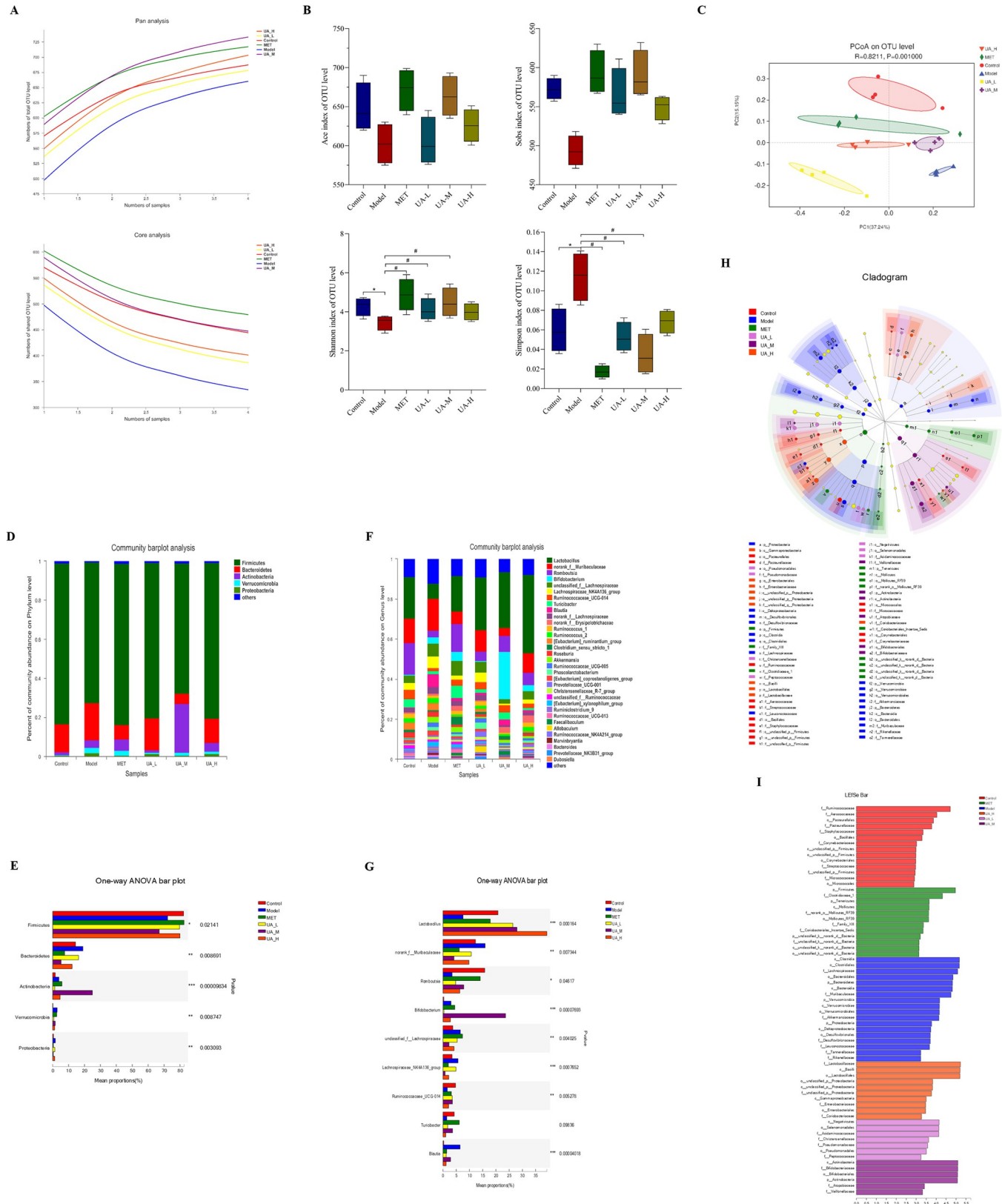

**Fig 2. UA treatment altered the specific microbiota population in T1DM rats.** To assess the effects of UA on gut microbiota, 16S rRNA sequencing of rat fecal DNA was performed. (A) Pan/core species analysis. The Pan/Core curve showed that the detection amount of each group is sufficient. (B) The alpha diversity analysis reflected the diversity and abundance of gut microbiota. (Chao index, Sobs index, Shannon index, and Simpson index). (C) Unweighted UniFrac-based principal coordinates analysis (PCoA) at the OUT level indicated the different β-diversity of gut microbiota. The ANOSIM analysis of variance confirmed the statistically significant separation of the six groups (R = 0.821, $P<0.05$). (D) The relative abundance of different bacterial phyla in each group. (E) A phylum-level differential analysis of these six groups. (F) Relative abundance of different bacterial genera in each group. (G) The differential analysis among the six groups at the genus level. (H) Linear discriminant analysis effect size (LEfSe) taxonomic cladogram depicting taxonomic association between microbiome communities from the six groups. Each node represents a specific taxonomic type. Yellow nodes denote the taxonomic features that are not significantly differentiated in each group. (I) Linear discriminant analysis (LDA) score derived from differentially abundant features in six groups. The LDA threshold was set to 2, and $P<0.05$ when LDA>2. (A-I) n = 4. Data are expressed as the mean ± SD. $**P<0.01$ versus the control, $*P<0.05$ versus the control; $\#\#P<0.01$ versus Model, $\#P<0.05$ versus Model.

significant differences in Chao and Sobs values among the groups. Therefore, the abundance of gut microbiota in each group was not affected. Principal co-ordinates analysis (PCoA) of the unweighted Unifrac distances was used to evaluate the differences in diversity of the intestinal microbiota among the samples and groups (Fig 2C). The results revealed that the distribution of species in the model and UA-L groups was separate from that in the control group, whereas the MET, UA-M, and UA-H groups were closer to the control group.

Next, we performed taxonomy-based analyses at the phylum and genus levels to evaluate the community composition of the gut microbes of rats in each group. The results showed that the composition of the gut microbiota in the UA-M and control groups was similar. The results also indicated that the gut microbiota included five major phyla: *Firmicutes, Bacteroidetes, Actinobacteria, Verrucomicrobia, and Proteobacteria. Firmicutes* and *Bacteroidetes* were the dominant flora in each group, accounting for more than 70% of the total. Compared to control rats, there was a reduction in *Firmicutes* and an increase in *Bacteroidetes* in model rats, but UA remarkably modulated the levels of *Firmicutes* and *Bacteroidetes* (Fig 2D and 2E). At the genus level, we found that the abundance of bacterial genera was significantly different among the six groups. The predominant bacterial genera in the UA and control groups were *Lactobacillus* and *Romboutsia*, whereas the abundance of *Lactobacillus* and *Romboutsia* declined in the model group. Meanwhile, *norank_f_Muribaculaceae* was the predominant bacterial genus in the model group, while the abundances of *norank_f_Muribaculaceae* was decreased in the UA and control groups (Fig 2F and 2G). Linear discriminant analysis (LDA) coupled with effect size measurements (LEfSe) was used to further identify the specific bacterial taxa that differed significantly in response to UA treatment. According to the analysis results, *Bacteroidetes, Verrucomicrobia* and *Proteobacteria* were the key bacterial types contributing to gut microbiota dysbiosis in the Model group. Nevertheless, *Firmicutes, Actinobacteria, Bifidobacteriales*, and *Lactobacillales* displayed relative enrichment in the UA and MET groups, which might be associated with the UA-mediated and MET- mediated alleviation of T1DM. (Fig 2H and 2I). The results demonstrated that T1DM led to a gut microbiota disorder, and UA treatment could alter gut microbiota diversity and composition.

## 3.3 UA treatment corrected the proportion of Th17 and Treg cells in rat with T1DM

To investigate the potential mechanism by which UA corrects Th17/Treg differentiation in T1DM rats, we assessed the differentiation of Th17/Treg cells in the spleen before and after UA intervention. As shown in Fig 3A and 3B, UA-treated rats had a significantly lower number of Th17 cells (Fig 3C) and a higher number of Treg cells than T1DM rats (Fig 3D). In addition, the proportion of Th17 and Treg cells in T1DM rats changed, and UA treatment alleviated this trend (Fig 3E). We also analysed the differentiation of Th17/Treg cells in mesenteric lymph nodes (S1 Fig). Strikingly, the differentiation of T cell responses in mesenteric

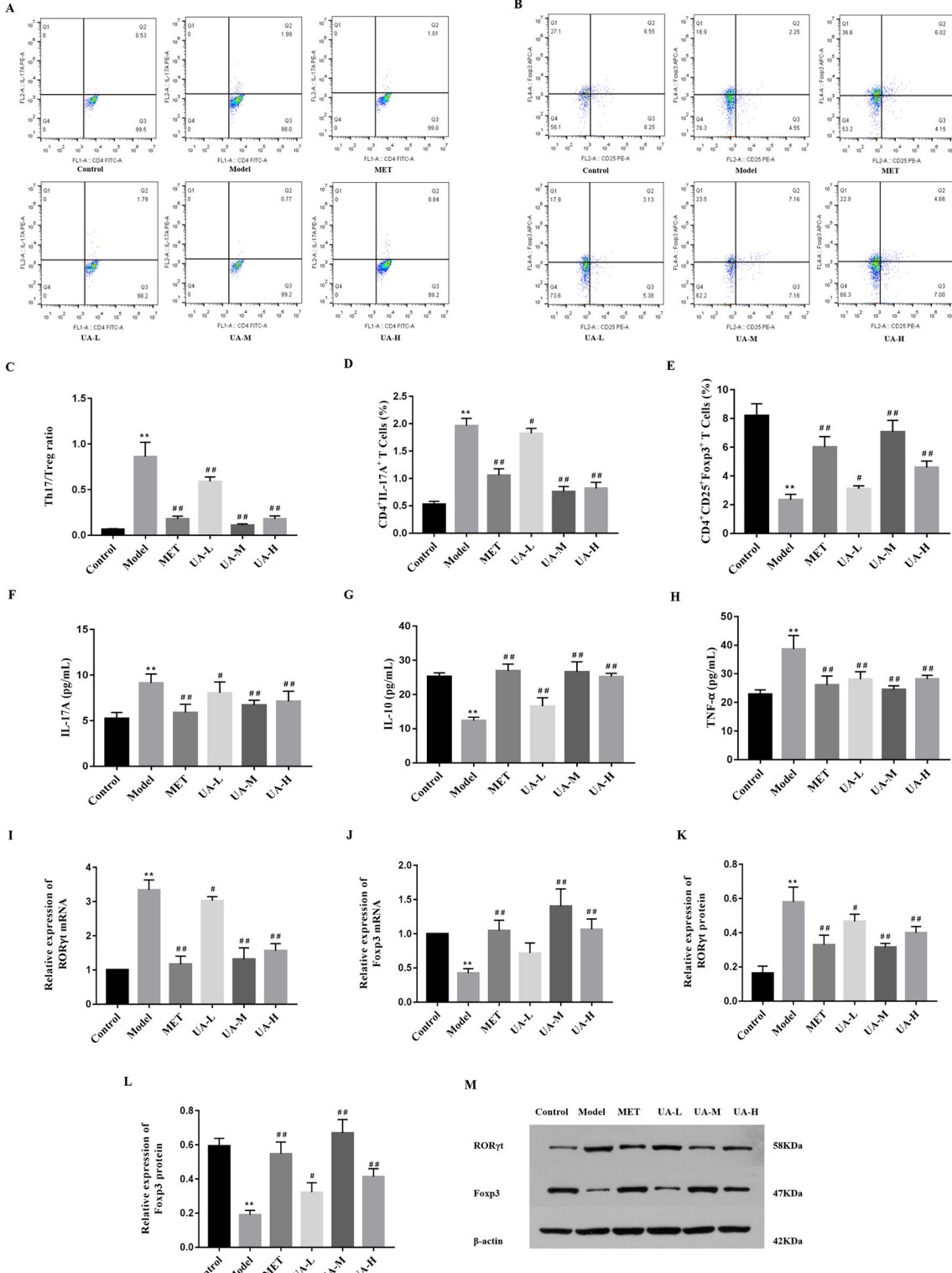

**Fig 3. UA treatment corrected the proportion of Th17 and Treg cells in rat with T1DM.** (A) CD4+IL-17A+ (Th17) cells and (B) CD4+CD25+Foxp3+ (Treg) cells in the spleen from the six groups were analyzed by flow cytometry. (C) The Th17/Treg ratio in the spleen. (D)The percentage of CD4+IL-17A+(Th17) cells in the spleen. (E) The percentage of CD4+CD25+Foxp3+ (Treg)cells in the spleens. (n = 6). ELISA was used to compare the levels of the cytokines IL-17A (F), IL-10 (G), and TNF-α (H) in blood serum between the six groups (n = 8). mRNA expression levels of RORγt (I) and Foxp3 (J) in the spleen of the rats were measured by real-time

quantitative PCR (n = 6). 419; (M) Western blotting results of RORγt, Foxp3, and β-actin in the spleen of T1DM rats (n = 3). The density of protein bands was quantified by software Alpha, which was normalized by comparison with β-actin and expressed as a percentage of the control. The data are expressed as the mean ± SD. **$P<0.01$ versus the control, *$P<0.05$ versus the control; ##$P<0.01$ versus Model, #$P<0.05$ versus Model.

lymph nodes manifested similar tendency in spleen, which indicated that UA treatment corrected the proportion of Th17 and Treg cells in rat with T1DM.

To confirm our flow cytometry results, we measured the serum levels of cytokines, such as IL-17A, IL-10, and TNF-α (Fig 3F–3H), as well as the mRNA and protein expression of Th17/Treg-associated genes such as RORγt and Foxp3, in T1DM rats (Fig 3I–3M). These results revealed that the levels of Foxp3 and IL-10 in control and UA rats were significantly higher than those in model rats, whereas the expression of RORγt, IL-17A, and TNF-α in UA rats was significantly lower than that in model rats. These results suggest that UA can correct the imbalance in Th17/Treg cell differentiation and regulate inflammatory factor secretion.

## 3.4 Associations between gut microbiota composition and T1DM phenotypes

RDA and correlation heat-map analyses were applied to investigate the correlations between the relative abundance of the different gut microbial communities and T1DM phenotypes (body weight, FBG, Th17 cells, Treg cells, IL-10, IL-17A, TNF-α, RORγt, and Foxp3). The results of the RDA analysis showed that Treg cells were most closely correlated with the distribution of the sample gut microbes (Fig 4A).

At the phylum level, we found that *Patescibacteria* and *Deferribacteres* were negatively correlated with Th17 cells and the Th17/Treg ratio but positively correlated with Treg cells. However, both *Verrucomicrobia* and *Cyanobacteria* exhibited a positive correlation with Th17 cells and the ratio of Th17/Treg (Fig 4B). At the genus level, *Lactobacillus* showed positive correlations with Treg cells, IL-10, and Foxp3 but were negatively correlated with IL-17A and RORγt. Seven genera including *Phascolarctobacterium*, exhibited a positive correlation with IL-17A, Th17cells, the ratio of Th17/Treg and RORγt, and a negative correlation with Treg cells and Foxp3. Six genera were positively associated with FBG but negatively associated with body weight (Fig 4C).

## 4. Discussion

T1DM is characterized by the immune-mediated depletion of pancreaticβ-cells, resulting in insulin deficiency and hyperglycaemia. MET can decrease glucose levels [28–30] and improve insulin sensitivity [31]. The American Diabetes Association has also made a positive recommendation for adding MET to insulin therapy in overweight patients with T1DM [32]. Therefore, MET was used as a positive control in our study.

UA is a natural pentacyclic triterpenoid derived from fruit and various traditional Chinese medicinal plants. It also possesses important antidiabetic effects, including lowering blood glucose levels and preserving pancreatic islet cell function [27, 33, 34]. For instance, UA exerts hypoglycemic effects by inhibiting the activities of α-amylase and α-glucosidase [35], activating TGR5 to enhance GLP-1secretion in type 1-like diabetic rats [36], and reducing the activity of protein tyrosine phosphatase 1 B [37]. Compared to MET, UA exhibits poor oral bioavailability, low solubility, and intestinal permeability [38]. However, metformin is associated with rare but serious events of lactic acidosis and gastrointestinal side effects [39, 40], whereas UA did not exhibit any significant toxicity in human THP-1 cells [41]. Moreover, a long-term toxicity study showed that oral dosing with UA for 90 consecutive days did not lead to toxic

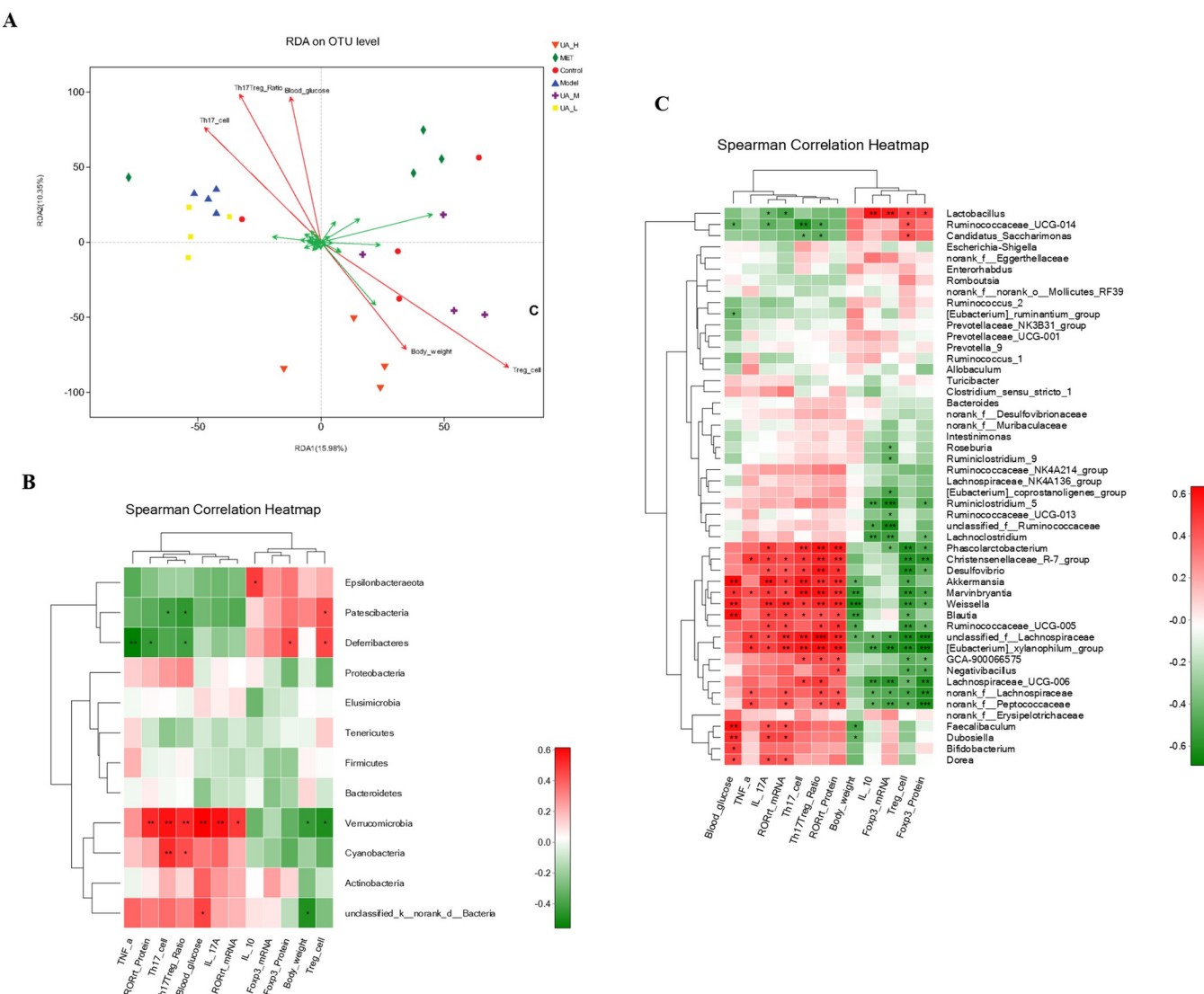

**Fig 4. Associations between gut microbiota composition and T1DM phenotypes.** (A) Redundancy analysis (RDA) based on microbial community and environmental factors. Green arrows represent species; red arrows represent environmental factors, and the length of the red arrows represents the degree of influence of environmental factors on species distribution. (B, C) The two-panel heatmap shows the correlations between phylum and genus- level flora and environmental factors. The corresponding R and P values were obtained by Spearman correlation analysis. The legend on the right shows the color range of different R values and set $|R| \geq 0.3$ as the screening threshold, *$P < 0.05$, **$P < 0.01$, ***$P < 0.001$.

effects at any dose. In contrast, recent studies have reported that gut microbiota dysbiosis and differentiation of Th17/Treg cells contribute to T1DM progression [11, 17, 42]. In addition, UA modulates the abundance and diversity of intestinal flora and inhibits Th17 cells differentiation [43, 44]. However, whether UA has similar protective and ameliorative effects in T1DM remains unknown.

In our study, we established a STZ-induced rat model of T1DM and found that UA can improve disease symptoms and pathological characteristics. In addition, our results preliminarily demonstrated that UA increased the diversity of the gut microbiota and corrected the imbalance of Th17/Treg cells in T1DM rats.

Increasing evidence indicates that the gut microbiota is involved in the occurrence and development of T1DM [45], and gut microbiota dysbiosis exists in patients with diabetes.

Fassatoui *et al.* studied the fecal bacterial composition of 31 participants, and the results showed that the proportions of *Firmicutes* and the ratio *Firmicutes/Bacteroidetes* decreased in participants with T1DM compared with those without diabetes [11]. Moreover, a study revealed that the gut microbiota of newly diagnosed T1D cases was altered, with higher levels of the genus *Bacteroides* compared with those of the control group [46]. By comparing the gut microbiota of Han Chinese patients with T1DM and those of healthy subjects, Huang *et al.* found that the gut microbiota of patients with T1DM was significantly different, with the *Bacteroides/Firmicutes* ratio increasing significantly [47]. Our study showed that the diversity and abundance of gut bacteria in T1DM rats were lower than those in the controls. Beta diversity analysis showed a reduction in *Firmicutes* and the ratio of *Firmicutes/Bacteroidetes* but an increase in *Bacteroidetes*, *Actinobacteria*, *Verrucomicrobia* and *Proteobacteria* in model rats. The above results also confirmed that there were certain similarities between the model rats used in our study and patients with T1DM. Interestingly, one study [48] demonstrated that specific inulin-type fructan fibers could delay the development of T1DM by modulating microbiota homeostasis. Zhao *et al.* reported that cinnamaldehyde might interfere with gut microbiota to modulate the expression levels of glucose metabolism-related genes, thus subsequently reducing blood glucose levels in T1DM mice [49]. In this study, we demonstrated for the first time that UA can increase the diversity and abundance of gut bacteria in T1DM rats. Specifically, UA treatment significantly increased the abundance of *Firmicutes* and the ratio of *Firmicutes* to *Bacteroidetes*, and significantly decreased the abundance of *Bacteroidetes*, *Actinobacteria*, *Verrucomicrobia*, and *Proteobacteria*. LEfSe analysis showed that UA mainly changed the composition of the gut microbiota by enriching *Bifidobacterium* and *Lactobacillus*. These findings suggest that UA may play an important role in restoring homeostasis of gut microbiota microecology in T1DM.

Among CD4⁺ cells, Th17 cells cause autoimmunity and inflammation, whereas Treg cells maintain immune tolerance to self-antigens and inhibit the inflammatory immune response [50]. They are considered to be the main effector subgroups involved in the pathogenesis of T1DM. Treg cells have been shown to be beneficial for the treatment of T1DM [51], and inhibiting Th17 cell differentiation may significantly delay the development of T1DM [52]. Ryba-Stanisławowska *et al.* found that the Treg/Th17 balance was disrupted in patients with T1DM, which may hasten the development of diabetic complications [53]. Ferraro *et al.* demonstrated that there was a Th17-cell bias and/or a Treg cell defect in patients with T1DM; thus, treatments targeted at suppressing Th17 cells and/or refitting Tregs may represent a solution for patients with diabetes [54]. Moreover, the differentiation of Th17 and Treg cells is directed by the master transcription factors RORγt and Foxp3. Th17 cells secrete the pro-inflammatory cytokine IL-17, which directly damages pancreatic β cells [17]. TNF-α is involved in the pathogenesis of T1DM by destroying pancreatic β-cell function, inhibiting insulin secretion, and inducing apoptosis [55]. In contrast, IL-10, an anti-inflammatory cytokine produced by Treg cells [56], inhibits the progression of diabetes in an animal model [57]. Our study showed that there was a significantly lower number of Treg cells and IL-10 but a higher number of Th17 cells, IL-17A, and TNF-α in T1DM rats than in controls. This finding is consistent with those of previous studies. Furthermore, Duan *et al.* found that metformin treatment significantly mitigated autoimmunity in NOD mice by inhibiting the differentiation of Th17 cells while promoting the development of Tregs [18]. Xu *et al.* reported that UA ameliorated the symptoms of experimental autoimmune myasthenia gravis by prompting Th17 cells to produce Th2 cytokines and up-regulate Treg cells [58]. Gudi *et al.* demonstrated that oral administration of high-purity yeast β-glucan suppressed insulitis by increasing the expression of IL10 and Foxp3 + T-cell frequencies in NOD mice [59]. Our study confirmed that UA dramatically inhibited Th17 cells, increased Treg cells, and improved the Th17/Treg balance in T1DM. Moreover,

Th17 cell markers, such as RORγt, IL-17A, and TNF-α, decreased, and Treg markers, such as FoxP3 and IL-10, increased. Our data indicated that UA could not only correct the imbalance of Th17/Treg cells, but also affect the differentiation of its cytokines to inhibit the inflammatory response, thus delaying the progression of T1DM.

The gut microbiome primarily influences the progression of immune diseases by regulating the Th17/Treg axis [60]. *Lactobacilli* and *Bifidobacteria* have been reported to be beneficial microbes, because they inhibit Th17 cell differentiation and increase Treg cell differentiation [61]. Th17 cells are produced by the colonization of *Bacteroides fragilis* [62], whereas Treg cell accumulation can also be induced by *Clostridia* [63]. Chen *et al.* found that both *C. butyricum* and norfloxacin treatment reduced Th17 cells and increased the percentage of Tregs, and gut microbiota alteration may be responsible for this effect [64]. Our study showed that UA treatment increased the levels of *Lactobacillus*, *Bifidobacterium*, and *Ruminococcaceae_UCG-014* but decreased the levels of *Bacteroidetes*. This indicates that UA modulates the abundance of the gut microbiota related to the differentiation of Th17 and Treg cells. However, the mechanism by which the gut microbiota modulates the differentiation of Th17/Treg cells during UA treatment is unclear and requires further investigation.

## 5. Conclusions

Our study demonstrated that UA alleviates the symptoms of STZ-induced T1DM in rats. UA treatment reduced FBG levels, delayed the progressive destruction of pancreatic β-cells, modulated intestinal microflora composition, and restored the Treg/Th17 cell balance. Collectively, these results suggested that UA is a promising drug for the prevention and treatment of T1DM. However, further research is required to confirm the positive results of the present study.

## Supporting information

**S1 Checklist.**
(PDF)

**S1 Fig. UA treatment corrected the proportion of Th17 and Treg cells in mesenteric lymph nodes of T1DM rats.** (A) CD4+IL-17A+ (Th17) cells and (B) CD4+CD25+Foxp3+ (Treg) cells in mesenteric lymph nodes from the six groups were analyzed by flow cytometry. (C) The percentage of CD4+IL-17A+(Th17) cells in mesenteric lymph nodes. (D) The percentage of CD4+CD25+Foxp3+ (Treg) cells in mesenteric lymph nodes. (E) The Th17/Treg ratio in mesenteric lymph nodes. (n = 6). The data are expressed as the mean ± SD. **P<0.01 versus the control, *P<0.05 versus the control; ##P<0.01 versus Model, #P<0.05 versus Model.
(TIF)

**S1 Raw images.**
(PDF)

## Acknowledgments

We thank Dr. Le Kang for the helpful discussions and critical reading of the manuscript.

## Author Contributions

**Conceptualization:** Xiaoli Zhang.

**Data curation:** ChaoJie Song, Yu Song.

**Formal analysis:** Weiwei Chen, Yingying Yu, Yang Liu, Xiaoli Zhang.

**Methodology:** Yingying Yu, HuanHuan Chen, Cong Tang.

**Project administration:** Yang Liu.

**Supervision:** Xiaoli Zhang.

**Writing – original draft:** Weiwei Chen.

**Writing – review & editing:** ChaoJie Song, Yu Song.

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
