## [Decision Letter · Decision Letter 0]

2 Aug 2022

PONE-D-22-16987Ursolic acid regulates gut microbiota and corrects the imbalance of Th17/Treg cells in T1DM ratsPLOS ONE

Dear Dr. Zhang,

Thank you for submitting your manuscript to PLOS ONE. After careful consideration, we feel that it has merit but does not fully meet PLOS ONE’s publication criteria as it currently stands. Therefore, we invite you to submit a revised version of the manuscript that addresses the points raised during the review process. It was felt that while the study has merit, it lacks conclusive and mechanistic experiments.  It was also felt that the authors need to discuss the rationale for use of ursolic acid and how it compares to metformin.  Many other important concerns including the language needs to be addressed by a point by point response to all the comments of the reviewers.

We look forward to receiving your revised manuscript.

Kind regards,

Pradeep Dudeja

Academic Editor

PLOS ONE

Journal Requirements:

2. As part of your revision, please complete and submit a copy of the Full ARRIVE 2.0 Guidelines checklist, a document that aims to improve experimental reporting and reproducibility of animal studies for purposes of post-publication data analysis and reproducibility: https://arriveguidelines.org/sites/arrive/files/Author%20Checklist%20-%20Full.pdf (PDF). Please include your completed checklist as a Supporting Information file. Note that if your paper is accepted for publication, this checklist will be published as part of your article.

5. PLOS requires an ORCID iD for the corresponding author in Editorial Manager on papers submitted after December 6th, 2016. Please ensure that you have an ORCID iD and that it is validated in Editorial Manager. To do this, go to ‘Update my Information’ (in the upper left-hand corner of the main menu), and click on the Fetch/Validate link next to the ORCID field. This will take you to the ORCID site and allow you to create a new iD or authenticate a pre-existing iD in Editorial Manager. Please see the following video for instructions on linking an ORCID iD to your Editorial Manager account: https://www.youtube.com/watch?v=_xcclfuvtxQ.

Reviewers' comments:

Reviewer's Responses to Questions

**Comments to the Author**

1. Is the manuscript technically sound, and do the data support the conclusions?

Reviewer #1: Yes

Reviewer #2: Yes

2. Has the statistical analysis been performed appropriately and rigorously? 

Reviewer #1: No

Reviewer #2: I Don't Know

3. Have the authors made all data underlying the findings in their manuscript fully available?

Reviewer #1: Yes

Reviewer #2: Yes

4. Is the manuscript presented in an intelligible fashion and written in standard English?

Reviewer #1: Yes

Reviewer #2: Yes

5. Review Comments to the Author

Reviewer #1: This manuscript by Chen et al. studies the potential of Ursolic acid to alter gut microbiota and correct Th17/Treg imbalance in Type 1 diabetes utilizing a rat model. The study is novel and supports the conclusions. There are several concerns;

Major concerns

1. The figures are difficult to follow, they are not labeled correctly and in some the statistics have not been conducted (Fig 1A,B,Fig 2B,C etc.) . Please make sure each figure is in one page with sub figures labeled accordingly. In addition, the figure legends should be very descriptive, and any abbreviation used (eg MET) should be defined in the legend as well.

2. Figure 2F is very wordy and the text is unclear even if the large image is downloaded. Please increase clarity of the figure.

3. The change in immune cells seems to be occurring in the spleen. Have the authors also looked at local tissue? Peyer’s patches, mesenteric lymph nodes and lamina propria? If not, could you speculate how gut microbiota can impact splenic Th17/Treg differentiation? What is the effect of UA?

4. All the beneficial effects of UA were similar to that of metformin an already well tolerated drug. Are there any side effects of UA? What is the benefit of using it over metformin? Please discuss

5. Please state molecular weight of proteins on the representative western blots

Minor concerns

1. The authors have mentioned that a recent study has shown there is an imbalance in Th17 and Tregs (this has been published in 2019) Is there any other recent study showing this?

2. Introduction is initiated with an abbreviation, please avoid doing so.

3. Scientific names should be in italics Page 4 line 90-91 (Salvia officinalis etc.) Please correct

Reviewer #2: In this manuscript, the authors describe their studies on the effects of ursolic acid (UA) in reversing gut microbiota dysbiosis and imbalanced Th17/Treg cells in T1DM rats. The experiments are technically well-designed and study results are appropriately described. However, rationale for choosing UA (in preference to many other phytochemicals showing hypoglycemic, antioxidant, anti-inflammatory, antitumor and neuroprotective properties) is not well addressed. Further, the studies described here are far from confirmatory to infer whether the ameliorating effects of UA on T1DM complications are direct effects of UA or mediated via correcting dysbiosis. This can be confirmed only by direct treatment with UA in vitro in cell culture experiments or in vivo in germ free animals.

There are multiple grammatical or typo errors that need attention (for example type I diabetes, not diabetic, in the Abstract). Also, the statements like “The results probably provided new research strategies for the prevention and treatment of T1DM” significantly dampens the scientific merit of the study.

Overall, the manuscript needs major revision with additional studies to confirm the claim that UA effects are via gut microbiota, not direct effects, or by downplaying the claim by revised writing of the Abstract, Results and Discussion/Conclusion sections.

6. PLOS authors have the option to publish the peer review history of their article (what does this mean?). If published, this will include your full peer review and any attached files.

Reviewer #1: No

Reviewer #2: **Yes: **Alip Borthakur, Ph.D.

---

## [Author Response · Author response to Decision Letter 0]

17 Sep 2022

Thank you very much for giving us an opportunity to revise our manuscript, we appreciate the reviewers very much for their positive and constructive comments and suggestions on our manuscript entitled “Ursolic acid regulates gut microbiota and corrects the imbalance of Th17/Treg cells in T1DM rats” (Manuscript Number: PONE-D-22-16987) by Weiwei Chen et al submitted to PLOS ONE. Those comments are all valuable and very helpful for revising and improving our paper, as well as the important guiding significance to our researches.And we tired our best to improve the manuscript and made some changes in the manuscript.All responses to reviewers and editors are included in a separate file labeled 'Response to Reviewers'. Finally, we would like to express our great appreciation to all comments on our paper. We hope that the manuscript will be accepted for publication in PLOS ONE.

---

## [Decision Letter · Decision Letter 1]

19 Oct 2022

Ursolic acid regulates gut microbiota and corrects the imbalance of Th17/Treg cells in T1DM rats

PONE-D-22-16987R1

Dear Dr. Zhang,

We’re pleased to inform you that your manuscript has been judged scientifically suitable for publication and will be formally accepted for publication once it meets all outstanding technical requirements and correction of minor spelling mistakes as outlined by one of the reviewer.

Kind regards,

Pradeep Dudeja

Academic Editor

PLOS ONE

Additional Editor Comments (optional):

Reviewers' comments:

Reviewer's Responses to Questions

**Comments to the Author**

1. If the authors have adequately addressed your comments raised in a previous round of review and you feel that this manuscript is now acceptable for publication, you may indicate that here to bypass the “Comments to the Author” section, enter your conflict of interest statement in the “Confidential to Editor” section, and submit your "Accept" recommendation.

Reviewer #1: All comments have been addressed

Reviewer #3: All comments have been addressed

2. Is the manuscript technically sound, and do the data support the conclusions?

Reviewer #1: Yes

Reviewer #3: Yes

3. Has the statistical analysis been performed appropriately and rigorously? 

Reviewer #1: Yes

Reviewer #3: Yes

4. Have the authors made all data underlying the findings in their manuscript fully available?

Reviewer #1: Yes

Reviewer #3: Yes

5. Is the manuscript presented in an intelligible fashion and written in standard English?

Reviewer #1: Yes

Reviewer #3: Yes

6. Review Comments to the Author

Reviewer #1: All comments have been addressed and is now suitable for publication.

Reviewer #3: Dear Authors

Even after professional editing, there are many mistakes in the manuscript. I have listed a couple of them.

Line 55: Spelling mistake: Druges should be drugs

Line 117: Bllank

Please check the consistency for faecal/fecal

Line 244: should be performed

7. PLOS authors have the option to publish the peer review history of their article (what does this mean?). If published, this will include your full peer review and any attached files.

Reviewer #1: No

Reviewer #3: No

---

## [Editor Report · Acceptance letter]

25 Oct 2022

PONE-D-22-16987R1 

Ursolic acid regulates gut microbiota and corrects the imbalance of Th17/Treg cells in T1DM rats 

Dear Dr. Zhang:

I'm pleased to inform you that your manuscript has been deemed suitable for publication in PLOS ONE. Congratulations! Your manuscript is now with our production department. 

Kind regards, 

on behalf of

Dr. Pradeep Dudeja 

Academic Editor

PLOS ONE